# Study on the Application of Kramers–Kronig Relation for Polyurethane Mixture

**DOI:** 10.3390/ma17122909

**Published:** 2024-06-14

**Authors:** Haisheng Zhao, Quanjun Shen, Peiyu Zhang, Zhen Li, Shiping Cui, Lin Wang, Wensheng Zhang, Chunhua Su, Shijie Ma

**Affiliations:** 1Key Laboratory of Highway Maintain Technology Ministry of Communication, Jinan 250102, China; zhaohaisheng@sdjtky.cn (H.Z.); zhangpeiyu@sdjtky.cn (P.Z.); cuishiping@sdjtky.cn (S.C.); wanglin@sdjtky.cn (L.W.); suchunhua@sdjtky.cn (C.S.); 2School of Highway, Chang’an University, Xi’an 710064, China; 3Shandong Expressway Group Innovation Research Institute, Jinan 250101, China; lingfei3651@sina.com; 4Shandong Tongda Luqiao Planning & Design Co., Ltd., Yantai 264119, China; lizhenokey@163.com; 5Wanhua Chemical Group Co., Ltd., Yantai 265599, China; wszhangb@whchem.com

**Keywords:** Kramers–Kronig relation, thermorheologically simple property, Hilbert integral transform, dynamic modulus, core–core, black space diagram

## Abstract

Polyurethane (PU) mixture, which is a new pavement mixture, exhibits different dynamic properties compared to a hot-mixed asphalt mixture (HMA). This paper analyzed whether the Kramers–Kronig (K–K) relation and thermorheologically simple properties applied to the PU mixture. Based on the results, the PU mixture exhibited thermorheologically simple properties within the test conditions. The time–temperature superposition principle (TTSP) was applicable for the PU mixture to construct a dynamic modulus master curve using the standard logistic sigmoidal (SLS) model, the generalized logistic sigmoidal (GLS) model, and the Havriliak–Negami (HN) model. The Hilbert integral transformed SLS and GLS models for the phase angle can accurately fit the measured phase angle data with newly fitted shift factors and predict the phase angle within the viscoelastic range. The core–core and black space diagrams both displayed single continuous smooth curves, which can be utilized to characterize the viscoelastic property of the PU mixture. The K–K relation is applicable for the PU mixture to obtain the phase angle master curve model, storage modulus, and loss modulus from the complex modulus test results with the test temperatures and loading frequencies. The phase angle of the PU mixture at extremely high or low test temperatures cannot be derived from the dynamic modulus data.

## 1. Introduction

Asphalt and asphalt mixtures are considered linear viscoelastic (LVE) materials that exhibit both elastic and viscous properties [1,2,3]. The mechanical behavior of asphalt and asphalt mixtures is dependent on temperature, loading frequency, loading history, and time. Capturing the viscoelastic behavior of asphalt and asphalt mixtures is crucial for understanding asphalt pavement performance [4,5,6]. Asphalt and asphalt mixtures in the LVE range typically exhibit thermorheologically simple (TRS) properties. Therefore, the time–temperature superposition principle (TTSP) can be applied to asphalt and asphalt mixtures. This implies that the behavior of asphalt and asphalt mixtures at high temperatures is equivalent to that at a low loading frequency and vice versa [5,6,7]. By applying the TTSP, the complex modulus data measured at different temperatures can be horizontally shifted along the frequency axis to a chosen reference temperature. This allows for the construction of a master curve to characterize the LVE properties of the asphalt binder or hot mixed asphalt mixture (HMA) [8,9,10]. The TTSP can be utilized to comprehend the viscoelastic properties of asphalt and asphalt mixtures across a broader loading frequency range than what is measurable [7,11,12]. The most prevalent method for characterizing the LVE behaviors of the asphalt binder or HMA involves measuring the complex modulus (dynamic modulus and phase angle) in the small strain region [9,12,13]. The dynamic modulus represents the stiffness of the asphalt and asphalt mixture, whereas the phase angle signifies the relative proportion of elastic and viscous response. A comparative study [14] has shown that the accuracy of a flexible asphalt pavement design and pavement performance prediction can be enhanced by incorporating a viscoelastic constitutive model that comprises dynamic modulus and phase angle master curves. For instance, in mechanistic–empirical pavement design (MEPDG), the dynamic modulus serves as a crucial input parameter for characterizing material properties in flexible asphalt pavement design [15].

The accuracy of phase angle measurement is crucial for calculating the elastic and viscous components. Determining the phase angle through temperature and frequency sweep tests in the laboratory is challenging, unreliable, and also prone to produce errors with high variability, due to sensitivity to signal noise and the need for accurate time-based data capture with current measurement technology [16]. Inaccurately measured phase angle data hinder application in further rheological and performance evaluations. Numerous research studies have utilized nonlinear minimization algorithms to construct the dynamic modulus master curve, often overlooking phase angle data [17].

For linear viscoelastic materials, the components of complex shear dynamic modulus can be shown in the rather simple Equations (1)–(4) in terms of the Kramers–Kronig (K–K) relations under certain conditions [18].
(1)$$G^{*}\left(i\omega \right)=G^{\prime }\left(\omega \right)+iG^{\prime \prime }\left(\omega \right)=G_{d}\left(\omega \right)exp\left(i\delta \left(\omega \right)\right)$$
(2)$$G^{\prime \prime }\left(\omega \right)≅\frac{\pi }{2}{\left(\frac{d{(G}^{\prime }\left(\mu \right))}{d(log\left(\mu \right))}\right)}_{\mu =\omega }$$
(3)$$G^{\prime }\left(\omega \right)-G^{\prime }\left(0\right)≅-\frac{\pi \omega }{2}{\left(\frac{d\left[\frac{G^{\prime \prime }\left(\mu \right)}{\mu }\right]}{d(log\left(\omega \mu \right))}\right)}_{\mu =\omega }$$
(4)$$\delta \left(\omega \right)≅\frac{\pi }{2}{\left(\frac{d(logG_{d}\left(\mu \right))}{d(log\left(\mu \right))}\right)}_{\mu =\omega }$$
where *μ* is the complex frequency, *ω* is the loading frequency, *δ* is the phase angle, *G*^*^ is complex shear modulus of viscoelastic material, *G_d_* is the equational deformation of *G*^*^ related to the phase angle *δ*.

The authors of [18] initially introduced the concept of predicting phase angle data based on the slope of dynamic modulus data relative to loading frequency, utilizing the generalized K–K relations [19]. The same concept was employed by the authors of [20] to acquire the phase angle data for the Christensen–Anderson model. Studies [16,21] have examined this concept for different asphalt mixtures, demonstrating high validity in predicting phase angle data from the slope of the dynamic modulus. The resulting measured and predicted phase angle data show similar master curve shapes. Based on the K–K relations, a study [22] utilized both the standard logistic sigmoidal (SLS) model and the generalized logistic sigmoidal (GLS) model to devise the corresponding mathematical function for the phase angle master curve. A study [15] compared the impact of aging on both the dynamic modulus and phase angle of HMA across nine aging levels. The results indicated that the phase angle master curve model, developed using the K–K relations, can accurately predict the phase angle data after aging with high precision. A study [23] was conducted to investigate the validity of the K–K relations for various modified and unmodified binders as well as asphalt mixtures. The authors of [24,25] adopted this approach to calculate both the phase angle and the horizontally shifted experimental data. The test results demonstrated that the dynamic modulus and phase angle data of the LVE materials could be linked using the K–K relations.

PU mixture is a new kind of pavement mixture that consists of PU binder, aggregate with certain gradation, and fillers. PU mixture exhibits higher performance compared with traditional hot-mixed asphalt mixture. The rheological property of PU mixture is uncertain, and it is urgent to analyze and demonstrate its rheological property. Based on the viscoelasticity principle and K–K relations, previous research has demonstrated the feasibility of predicting the phase angle master curve using the corresponding dynamic modulus data. The objective of this paper is to verify if the K–K relations apply to the PU mixture, in other words, if only dynamic modulus data of the PU mixture are available from different test methods (e.g., fatigue test and indirect tensile test), whether phase angle data can be obtained from corresponding dynamic modulus data in terms of the K–K relations. The following studies were performed: (a) to use SLS, GLS, and Havriliak–Negami (HN) [26] models to construct the dynamic modulus master curve, combined with the Williams–Landel–Ferry (WLF) shift factor equation [21], and assess if the TTSP applied for the PU mixture; (b) to perform a Hilbert integral transform [19] on the SLS, GLS, and HN models to build the corresponding phase angle master curve function models and use these models to construct the phase angle master curves to verify if the transformed models can predict the phase angle using the slope of dynamic modulus; (c) to use core–core and black space diagrams to access the precision of phase angle prediction.

## 2. Materials and Methods

### 2.1. Material and Test

In this study, the basalt aggregate and a single-component and wet-setting type polyurethane provided by Wanhua Chemical Group Co., Ltd. (Yantai, China) were utilized to fabricate the PU mixture. The gradation of the aggregate is listed in Table 1. The optimal ratio of PU binder content to aggregate was determined to be 5.0% through the Marshal test.

The specimens of the PU mixture were created using the Superpave gyratory compactor (SGC) with a size of 175 mm in height and 150 mm in diameter. After curing, the specimens were cut into a size of 150 mm in height and 100 mm in diameter. The dynamic modulus test was conducted following the AASHTO: TP–79 (2010) procedure using the Asphalt Mixture Performance Tester (AMPT) in load-controlled uniaxial compression mode. The strain range for the dynamic modulus test was maintained from 75–125 με, ensuring that the test response remained within the linear viscoelastic range. The recommended test condition encompassed six temperatures (5 °C, 15 °C, 25 °C, 35 °C, 45 °C, and 55 °C) and nine loading frequencies (25, 20, 10, 5, 2, 1, 0.5, 0.2, and 0.1 Hz). Eight duplicates were utilized for calculating the average values of dynamic modulus and phase angle data that would be used for analysis purposes. Test results were divided into two groups and each group consisted of average values of four duplicates. The first group was used as a control group and the second group was used as a verifying group.

### 2.2. Master Curve Models

#### 2.2.1. Dynamic Modulus Master Curve Models

The standard logistic sigmoidal (SLS) model [27], which is defined in Equation (5), was introduced to (1) describe the linear viscoelastic behaviors of HMA across different temperatures and frequencies, (2) fit measured dynamic modulus data, and (3) establish the dynamic modulus master curve.
(5)$$log(\left|E^{*}\right|)=\delta +\frac{\alpha }{1+e^{\beta +\gamma \cdot \log\left(f_{r}\right)}}$$
where |*E^*^*|, dynamic modulus, MPa; *f_r_*, load frequency at the reference temperature, Hz; *δ*, *α*, *β*, and *γ*, fitting parameters. *δ* and *δ* + *α* represent the minimum and maximum values of *E^*^*; *β* and *γ*, shape parameters. Equation (5) follows a standard log function format, leading to a symmetrical curve for the evolution of dynamic modulus [27,28,29,30].

The SLS model exhibits a symmetrical shape and offers a good fit for symmetric measured data, but it cannot fit nonsymmetric curves with acceptable accuracy. Due to the random heterogeneous characteristic of HMA [31,32], its dynamic modulus frequently displays an asymmetric pattern. Consequently, the generalized logistic sigmoidal (GLS) model, as seen in Equation (6), was introduced by [33] to more effectively capture and fit the asymmetric evolution of dynamic modulus data. A generalized version of the standard logistic sigmoidal function can be used to fit asymmetric curves.
(6)$$\log(\left|E^{*}\right|)=\delta^{\prime }+\frac{\alpha^{\prime }}{[1+\lambda \cdot e^{\beta^{\prime }+\gamma \cdot log\left(f_{r}\right)}{]}^{\frac{1}{\lambda }}}$$
where *δ*′, *α*′, *β*′, and *γ*′, fitting parameters; *λ*, added parameter for the asymmetric shape of the function; *δ*′ and *δ*′ + *α*′ represent the minimum and maximum value of *E^*^*; *β*′ and *γ*′, shape parameters of the GLS model.

Tschoegl [19] introduced the original HN model [26] to model the complex modulus of HMA. The modified HN model is expressed in Equation (7) and includes additional parameters to better fit experimental data.
(7)$$E^{*}\left(\omega \right)=\left|E_{0}\right|+\frac{\left|E_{\infty }\right|-\left|E_{0}\right|}{[1+{\left(\frac{\omega_{0}}{i\cdot \omega }\right)}^{\alpha^{\prime \prime }}{]}^{\beta^{\prime \prime }}}$$
where *i*^2^ = −1; *ω*_0_ = 1/*τ*_0_; *ω* is the angular frequency; *τ*_0_ determines the horizontal position of the real or imaginary part of the dynamic modulus along the frequency axis; |*E_∞_*| and |*E*_0_| represent the dynamic modulus as *ω* approaches ∞ and 0, respectively; *α*″ and *β*″ control the width and skewness of the curve, respectively.

#### 2.2.2. Integral Transform

The Hilbert integral transforms [19] can be used to translate between the dynamic modulus and phase angle, or the storage modulus and loss modulus, also known as the K–K relations. Theoretically, the mathematical model for the dynamic modulus or storage modulus can be transformed into models for the phase angle through the Hilbert integral transform. The transformation process is defined in Equation (8) [19].
(8)$$\varphi \left(\omega \right)=\frac{\pi }{2}\cdot \frac{d(log\left(\left|E^{*}\left(\omega \right)\right|\right))}{d(log\left(\omega \right))}$$

Using Equation (4), the SLS, GLS, and HN models in Equations (5)–(7) can be transformed into corresponding mathematical models for the phase angle, which are shown in Equations (9)–(11).
(9)$$\varphi \left(\omega \right)=-\frac{\pi }{2}\cdot \frac{\alpha \cdot \gamma \cdot e^{\beta +\gamma \cdot log\left(\omega \right)}}{{\left(1+e^{\beta +\gamma \cdot log\left(\omega \right)}\right)}^{2}}$$
(10)$$\varphi \left(\omega \right)=-\frac{\pi }{2}\cdot \frac{\alpha^{\prime }\cdot \gamma^{\prime }\cdot e^{\beta^{\prime }+\gamma^{\prime }\cdot log\left(\omega \right)}}{{\left(1+\lambda \cdot e^{\beta^{\prime }+\gamma^{\prime }\cdot log\left(\omega \right)}\right)}^{\frac{1}{\lambda }+1}}$$
(11)$$\tan\varphi \left(\omega \right)=\frac{\sin\left(\frac{\alpha^{\prime \prime }\cdot \pi }{2}\right)}{{\left(\frac{\omega }{\omega_{0}}\right)}^{\alpha^{\prime \prime }}+\cos\left(\frac{\alpha^{\prime \prime }\cdot \pi }{2}\right)}$$

### 2.3. Shift Factor

According to the TTSP, the isotherms of *E^*^* versus reduced frequency at different test temperatures could be horizontally shifted to the selected referenced temperature *T*_0_ to create a smooth dynamic modulus master curve for assessing the LVE properties of asphalt and asphalt mixtures.

Where log(*f*), frequency in experiment temperature; log(*f_r_*), reduced frequency in reference temperature; *α_T_*, shift factor. The shift factor is determined by the shifted distance of different isotherms and is given in Equation (12).
(12)$$log\left(f_{r}\right)=log\left(f\right)+log\left(\alpha_{T}\right)$$

The temperature shift factor *α_T_* (at temperature *T*) can be determined using the Williams–Landel–Ferry (WLF) [10,23,34,35] empirical equations for HMA, which are most accurate at temperatures close to the glass transition temperature. The equation is shown in Equation (13).
(13)$$log\left(\alpha_{T}\right)=\frac{-C_{1}\cdot \left(T-T_{0}\right)}{C_{2}+\left(T-T_{0}\right)}$$
where *C*_1_ and *C*_2_, constants; *T*, test temperature; and *T*_0_, reference temperature, set as 20 °C in this paper.

### 2.4. Nonlinear Regression Procedure

The process of fitting the master curve model involves regressing the parameters of the model, which are then used to construct the master curve for describing the LVE properties. A nonlinear regression procedure is adopted to minimize the difference between predicted and measured values, as shown in Equations (14) and (15). The Excel solver was used in the nonlinear regression procedure.
(14)$$E_{Error}^{*}=\sum_{i}^{i=n}{\left(logE_{Exp}^{*}\left(\omega_{i}\right)-logE_{Prd}^{*}\left(\omega_{i}\right)\right)}^{2}$$
(15)$$\delta_{Error}=\sum_{i}^{i=n}{\left(\delta_{Exp}\left(\omega_{i}\right)-\delta_{Pre}\left(\omega_{i}\right)\right)}^{2}$$
where *E^*^_Exp_*, experimental dynamic modulus; *E^*^_Pre_*, predicted dynamic modulus using different mathematical models; *δ_Exp_*, experimental phase angle; *δ_Pre_*, predicted phase angle using different mathematical models.

Four indexes were introduced to evaluate the fitting accuracy, including *S_e_*/*S_y_* (ratio of the standard error (*S_e_*) to the standard deviation (*S_y_*)) minimization, *R*^2^ maximization, sum of square error (*SSE*) minimization, and Error^2^ minimization, shown in Equations (16)–(20).
(16)$$S_{e}=\sqrt{\frac{1}{n-p-1}\cdot \sum_{i}^{n}\left|{\hat{x}}_{i}-x_{i}\right|}$$
(17)$$S_{y}=\sqrt{\frac{1}{n-1}\cdot \sum_{i}^{n}\left|{\hat{x}}_{i}-{\bar{x}}_{i}\right|}$$
where *x_i_* is the measured dynamic modulus, ${\hat{x}}_{i}$ is the predicted dynamic modulus, and ${\bar{x}}_{i}$ is the mean value of the measured dynamic modulus.
(18)$$R^{2}=1-\frac{\left(n-p-1\right)\cdot S_{e}^{2}}{\left(n-1\right)\cdot S_{y}^{2}}$$
where *n* is the sample size and *p* is the number of parameters to be estimated.
(19)$$SSE=\sum_{i}^{i=n}\frac{{\left(E_{Exp}^{*}\left(\omega_{i}\right)-E_{Prd}^{*}\left(\omega_{i}\right)\right)}^{2}}{{\left(E_{Exp}^{*}\left(\omega_{i}\right)\right)}^{2}}$$
(20)$${Error}^{2}=\sum_{i}^{i=n}{\left(E_{Exp}^{*}\left(\omega_{i}\right)-E_{Prd}^{*}\left(\omega_{i}\right)\right)}^{2}$$

## 3. Results

### 3.1. Viscoelastic Properties

The measured dynamic modulus and phase angle data, obtained under various test temperatures and loading frequencies, are presented in Figure 1 to assess the influence of test temperature and loading frequency on the dynamic properties of the PU mixture.

### 3.2. Dynamic Modulus Master Curve

The SLS, GLS, and HN models were employed along with the WLF function to fit the dynamic modulus results using nonlinear regression. The parameters obtained from these mathematical models were introduced to build the corresponding dynamic modulus master curves. The fitting accuracy results for each model are listed in Table 2. The isotherms of measured dynamic modulus data at various test temperatures were shifted to the master curve using shift factors corresponding to those test temperatures. The shifted results are also presented in Figure 2. Figure 3a displays all three dynamic modulus master curves for the SLS, GLS, and HN models for comparison. Figure 3b presents the dynamic modulus master curves with shifted measured dynamic modulus data. Figure 3c compares the measured dynamic modulus with the predicted dynamic modulus obtained from three different models.

### 3.3. Phase Angle Master Curve

Theoretically, the phase angle master curve model equation can be derived from the corresponding dynamic modulus master curve model equation in terms of the K–K relation as shown in Equations (5)–(7). The fitted parameters of the dynamic modulus master curve models were substituted in Equations (5)–(7) to obtain predicted phase angle data. The predicted phase angle data results are shown in Figure 4.

Theoretically, the shift factors for the dynamic modulus and phase angle master curves were the same. The corresponding phase angle master curve models, using the WLF shift factor results obtained in Section 3.2, were combined to fit the measured phase angle data. The fitting parameters were then introduced to construct the corresponding master curves. The master curves for different models with the same shift factors of the dynamic modulus are plotted in Figure 5a, Figure 6a, and Figure 7a, respectively. The master curves with newly fitted shift factors for different models are plotted in Figure 5b, Figure 6b, and Figure 7b, respectively. The master curves constructed using different shift factors are plotted together with shifted measured phase angle data in Figure 5c, Figure 6c, and Figure 7c, respectively. The fitting accuracy results for different models are listed in Table 3 and Table 4. Then, the isotherms of measured phase angle data at different test temperatures were shifted to the master curve using the shift factors at corresponding test temperatures. All three phase angle master curves for the SLS, GLS, and HN models, with shifted measured phase angle data, are plotted together for comparison in Figure 8a. The shift factors for different models are shown in Figure 8b. In Figure 8c, the measured phase angle data are plotted against the predicted phase angle data by three different models. The dynamic modulus test data were divided into two groups, the first group was used to fit the master curve parameters and to construct the corresponding master curves and the second group was used as the control group to verify the models constructed by the first group. The phase angle data were plotted against the predicted phase angle data at the same loading frequency and test temperature to verify the precision of the SLS and GLS models, and the comparison results are shown in Figure 8d.

### 3.4. Core–Core and Black Space Diagrams

The plot of the loss modulus as a function of the storage modulus, known as the core–core, is shown in Figure 9 for both the SLS and GLS models. The black space diagrams, which represent the plot of the dynamic modulus as a function of the phase angle, are presented in Figure 10 for both the SLS and GLS models.

## 4. Discussion

### 4.1. Viscoelastic Properties

The mechanical property of the LVE material is dependent on test temperature, loading frequency, and loading history [16]. The complex dynamic modulus, including dynamic modulus and phase angle, can be adapted to represent the mechanical property (relationship between stress and strain) which depends on test temperature and loading frequency for LVE materials. Furthermore, the LVE material exhibits thermorheologically simple properties within the LVE range. The dynamic modulus of HMA, a typical thermorheologically simple material, increases with loading frequency and decreases as the test temperature increases. Conversely, the phase angle decreases with loading frequency and increases with test temperature, reaching peak values. Interpretation of the relationship between test temperature and loading frequency for LVE materials such as HMA enables the identification of identical mechanical behavior under various experimental conditions. For instance, the complex modulus measured at low loading frequency (low test temperature) is comparable to that measured at high temperature (high loading frequency).

Figure 1 presents the laboratory-measured dynamic modulus and phase angle isotherms at test temperatures ranging from 5 °C to 55 °C and loading frequencies of 25 Hz, 20 Hz, 10 Hz, 5 Hz, 2 Hz, 1 Hz, 0.5 Hz, 0.2 Hz, and 0.1 Hz to demonstrate the influence of test temperature and loading frequency.

As shown in Figure 1a, there was a clear pattern in the dynamic modulus of the PU mixture as the loading frequencies and test temperatures varied. At constant loading frequencies, the dynamic modulus decreased as the test temperature increased. The rate of decrease gradually increased from 43% to 50.1% as the loading frequency decreased from 25 Hz to 0.1 Hz. At constant test temperatures, the dynamic modulus increased as the loading frequency increased. the rate of increase ranged from 2.36% to 32.9% as the test temperature increased from 5 °C to 55 °C.

As shown in Figure 1b, the phase angle of the PU mixture also displayed a clear trend of change with increasing loading frequencies and test temperatures. At constant loading frequency, the phase angle steadily increased with rising test temperature, and the rate of increase rose from 35.4% to 44% as the loading frequency decreased from 25 Hz to 0.1 Hz, with the highest value observed at a loading frequency of 0.5 Hz. At constant test temperature, the phase angle decreased as the loading frequency increased from 0.1 Hz to 25 Hz. Compared to HMA, the phase angle of the PU mixture exhibited a changing pattern, with a steady increase or decrease in response to increases in test temperature or loading frequency. The phase angle of HMA displays peaks at high test temperatures, and the changing trend varies depending on the test temperature. As the test temperature increased from 5 °C to 55 °C, the decreasing rate ranged from 24.1% to 41.3%, indicating that it did not change gradually with each increase in temperature. The highest value was observed at a test temperature of 25 °C.

Based on the foregoing discussion, the impact of test temperatures and loading frequencies on the dynamic modulus and phase angle of the PU mixture is evident. As test temperatures and loading frequencies rose, dynamic modulus and phase angle underwent gradual alterations. This result aligns with the characteristics of thermorheologically simple materials, indicating that the PU mixture demonstrates thermorheologically simple behavior. Nevertheless, the rates of change for dynamic modulus and phase angle exhibited varying patterns. In dynamic modulus test data, the rate of change in test temperature and loading frequency progressed steadily with increments in test temperature or loading frequency. Conversely, for phase angle, the rate of change did not progress steadily with rising test temperature or loading frequency. This suggests that the PU mixture’s dynamic modulus is less susceptible to variations in test temperature and loading frequency, and the test precision of dynamic modulus surpasses that of phase angle.

### 4.2. Dynamic Modulus Master Curve

If a thermorheologically simple material like HMA is tested within the LVE range, the TTSP, which reflects the equivalence between time and temperature, can horizontally shift the test dynamic modulus isotherms along the frequency axis to a preselected reference temperature *T*_0_ at a reduced frequency [5,6,7]. The shifted test dynamic modulus isotherms at different test temperatures can construct a single continuous smooth master curve at the selected reference test temperature to fully characterize the LVE behavior of HMA [9]. The amount of horizontal shift multiplied by the loading frequency for each isotherm is called the time–temperature shift factor α_T_, which can establish an equivalence between the LVE properties measured at a frequency *f* versus a temperature *T* and a reduced frequency *f*_0_ versus a reduced temperature *T*_0_. The dynamic modulus master curve, combined with the shift factor, can predict the LVE behavior of HMA in a loading frequency and test temperature range broader than those used in experimental conditions [7]. This is necessary because asphalt pavement experiences a wide range of temperature conditions in the field. This method is commonly applied to both polymers [7] and asphalt mixtures [8,10].

The dynamic modulus isotherms, obtained at various test temperatures, are shifted to a lower frequency to establish the master curve corresponding to a reference temperature of 20 °C. This process employs the SLS, GLS, and HN models, which are depicted in Figure 2. The shifted factors are plotted in Figure 3b.

The PU mixture’s dynamic modulus decreases as the test temperature increases, as shown in Figure 1. The tested temperature range was not wide enough to analyze the properties of the PU mixture at an extremely high temperature, which was limited by the experimental equipment. Whereas the master curve can predict the dynamic modulus at a specific loading frequency, and the properties of the LVE material can vary with the interrelationship between loading frequency and temperature. As detailed in Figure 2d, when the reference temperature changed from 20 °C to 15 °C (or 25 °C), the master curve of the SLS model shifted only to the left (or right). Therefore, the choice of reference temperature does not impact the shape of the master curve, and the dynamic modulus master curve at a specific reference temperature was adopted to predict the dynamic modulus at various test temperatures with the shifted factors. The dynamic modulus master curves obtained using the SLS, GLS, and HN models exhibited an “S” shape as a function of loading frequency (in log form, within the range of 10 × 10^−10^~10 × 10^10^ rad/s). As shown in Figure 3a, all three models’ master curves exhibited similar shapes, with those of the SLS and GLS models nearly identical except at high loading frequency. The HN model’s master curve showed lower values compared to the SLS and GLS models. The GLS model exhibits the highest values.

The dynamic modulus of the thermorheologically simple material remains at extremely low frequencies equal to that at high test temperatures. As depicted in master curves in Figure 2, the dynamic modulus values for the PU mixture did not approach zero at extremely high temperatures, unlike HMA. This indicates that the PU binder retains its viscous property even at extremely high temperatures and would still be subject to external pressure combined with the aggregate skeleton. The PU mixture remains in the PU binder-dominated phase across a wider range of reduced frequency and does not shift to an aggregate-dominated phase, even at low reduced frequency or high temperature. The result indicates that the PU mixture offers improved resistance to deformation at high temperatures compared to HMA, given that the dynamic modulus of the asphalt approaches zero at extremely high temperatures, as documented in the literature [35].

The fitting accuracy of the three models was similar, with the SLS and GLS models achieving equal accuracy as shown in Table 2. The indexes of the goodness of fitting (SSE and *S_e_*/*S_y_*) for the HN model were bigger than those of the SLS and GLS models, while the Error^2^ index of the HN model was slightly lower. Based on the fitting results, all three models had relatively small values for the indexes of the goodness of fitting and were relatively close. Due to the similarity in fitting accuracy, it is challenging to solely rely on these indexes to determine the goodness of fitting. Hence, the graphical visualization analysis was introduced to compare the fitting result as shown in Figure 2a–c and Figure 3c. The lab-measured dynamic modulus data can be shifted to the master curve by combining the corresponding shift factors, as illustrated in Figure 2a–c. It can be inferred from Figure 2a,b that the shifted dynamic modulus data for the SLS and GLS models were slightly offset from the master curves, which were distributed on the right side. In Figure 2c, the shifted dynamic modulus data of the HN model closely follow the master curve. Based on these observations, the HN model exhibits superior precision in fitting the laboratory-measured dynamic modulus of the PU mixture compared to both the SLS and GLS models. The predicted dynamic modulus values from three models were plotted against the corresponding measured dynamic modulus data in Figure 3c. All the predicted dynamic modulus points except for two, which were generated by the HN model, were distributed along the line of equality. This suggested that the three models had a high level of precision when predicting the dynamic modulus. The linear fitting procedure was used to match the predicted and measured dynamic modulus data points, and the result is listed in Table 5. Based on the data in Table 5, the three models had similar fitting equations, but the *R*^2^ values for the SLS and GLS models were higher than those of the HN model. Drawing from the comparison and analysis above, all three models demonstrated high accuracy in predicting the dynamic modulus data of the PU mixture and can be utilized to predict and assess its dynamic properties. Meanwhile, the HN model is capable of constructing a master curve that aligns with the shifted measured dynamic modulus data.

The shift factor, log(*α_r_*), indicates the difference between two temperatures. The values of log(*α_r_*) across various test temperatures, which shape the master curve, reveal the temperature-dependent nature of the LVE material. The shift factors of the SLS and GLS models were closely aligned and slightly larger than those of the HN model. This indicates that the PU mixture exhibits consistent temperature dependence that is not influenced by the model type. Hence, the selected WLF function is suitable for modeling the frequency dependence of the mechanical properties of the PU mixture.

Based on the discussion above, the shift factors used to shift the dynamic modulus isotherms tested at various temperatures for the PU mixture can be determined by the WLF function. This suggests that the TTSP method is suitable for shifting the measured dynamic modulus data for the PU mixture. The SLS, GLS, and HN models can be used to construct a single master curve for predicting the dynamic modulus data of the PU mixture. The shapes of these master curves across different models remain similar, indicating that they were not significantly affected by the selected reference temperatures. Therefore, the dynamic modulus of the PU mixture reflects its frequency and temperature dependence, which is typical for viscoelastic materials. This indicates that the PU mixture exhibits certain viscoelastic properties.

### 4.3. Prediction of the Phase Angle Master Curve from Corresponding Dynamic Modulus Master Curve

This section evaluates the utilization of a fundamental relationship method to predict phase angle based on the slope of the log–log of the dynamic modulus master curve, also known as the K–K relations. Equations (5)–(7) were utilized to construct the phase angle master curves for the SLS, GLS, and HN models, incorporating the WLF shift factors. The equations were derived from the corresponding dynamic modulus in terms of the K–K relations. Therefore, it is expected that Equations (5)–(7) should utilize the fitted parameters obtained from the dynamic modulus master curve fitting process. The corresponding phase angle master curves are presented in Figure 4a. The predicted phase angle data are plotted versus the laboratory-measured phase angle data in Figure 4b.

According to Figure 4a, the predicted phase angle master curve of all three models showed small values, with peak values ranging from 5~10°. The maximum laboratory-measured phase angle was approximately 9° at a test temperature of 55 °C. As illustrated in Figure 1b, the phase angle increases as the test temperature rises. The phase angle values of the PU mixture will exceed 9° when the test temperature exceeds 55 °C. This indicates that the phase angle master curves constructed using the fitting parameters of the dynamic modulus master curve do not apply to the PU mixture. The predicted phase angle values from different models were distributed along the line of equality but with large discrepancies. Most of the predicted phase angle data were smaller than the measured phase angle data. This further confirms that the phase angle master curve with the fitted parameters of the dynamic modulus does not apply to the PU mixture. The phase angle master curve models should be fitted with new parameters to accurately characterize the LVE properties of the PU mixture.

In theory, the shift factors used to shift the phase angle and dynamic modulus data should be identical. The phase angle master curves constructed using the SLS, GLS, and HN models combined with the WLF shift factors of the corresponding dynamic modulus are displayed in Figure 5a, Figure 6a, and Figure 7a, respectively. A separate phase angle master curve fitting procedure was conducted based on Equations (5)–(7) with newly fitted WLF shift factors. The results are presented in Figure 5b, Figure 6b, and Figure 7b, respectively. The experimental phase angle data were shifted to the master curve at the reference temperature of 20 °C for a visual comparison between measured and predicted values.

For the SLS model, the SLS–1 model represents the phase angle master curve constructed using the same WLF shift factors as the dynamic modulus, while the SLS–2 model represents the phase angle master curve constructed using newly fitted WLF shift factors. The master curves of the SLS–1 model and SLS–2 model displayed similar shapes, as shown in Figure 5c. As the loading frequency increased, the phase angle gradually decreased. There was no peak on the phase angle master curves, indicating that no phase transition occurred during the change in loading frequency (or test temperature). This indicates that the PU mixture is dominated by both the PU binder and aggregate, which differs from HMA [36,37]. The result of the SLS–2 model was higher than that of the SLS–1 model when the loading frequency was less than 10 rad/s. However, both master curves merged when the loading frequency exceeded 10 rad/s, as shown in Figure 5c. The measured phase angle data were shifted to the master curves using corresponding WLF shift factors. The shifted phase angle data points were distributed along the master curve. When compared to the SLS–2 model, the shifted phase angle data points of the SLS–1 model exhibited slighter dispersion. This was further supported by comparing measured and predicted phase angle data in Figure 5d. The data points of the SLS–1 model exhibited a larger dispersion compared to those of the SLS–2 model. The linear fitting process was carried out for the predicted and measured phase angle data points shown in Figure 5d, and the fitting results are listed in Table 6.

Based on the fitting accuracy presented in Table 3, the SLS–2 model produced higher predicting accuracy with higher *R*^2^ values and lower values of SSE, Error^2^, and *S_e_*/*S_y_*. As shown in Table 6, the SLS–2 model can predict more accurate phase angle data compared to the SLS–1 model. This is because the linear equation fit for the SLS–2 model resulted in a smaller intercept, a larger slope of the line, and a higher *R*^2^ value. Therefore, the SLS–2 model is capable of fitting and predicting the measured phase angle with higher precision compared to the SLS–1 model.

For the GLS model, the GLS–1 model and GLS–2 model represent the phase angle master curve constructed using the same WLF shift factors of corresponding dynamic modulus and newly fitted WLF shift factors, respectively. The master curve of the GLS–1 model and GLS–2 model exhibited similar shapes, with the phase angle decreasing steadily as the loading frequency increased. There were also no peaks observed on the master curves of the GLS–1 model and GLS–2 model, indicating that the PU binder did not undergo any phase transitioning within the loading frequency range. As depicted in Figure 6c, the master curve of the GLS–2 model was higher than that of the GLS–1 model. When the loading frequency exceeded 10 rad/s, the master curves of GLS–1 and GLS–2 merged. The measured phase angle data points were shifted onto the master curve using different WLF shift factors, and the shifted phase angle data points were distributed along the master curves. Additionally, as shown in Figure 6a,b, the shifted phase angle points for the GLS–2 model exhibited small dispersion compared to those of the GLS–1 model, which was also evident from Figure 6d. In Figure 6d, both the measured and predicted phase angle data points for the GLS–1 model and GLS–2 model are plotted, with the data points for GLS–1 exhibiting greater dispersion than those of GLS–2. The linear relationship was fitted between the experimental and predicted values obtained from different predictive models. The fitted equations were then used to compare on a unity plot (x = y) in Figure 6d. The fitting results are listed in Table 7.

As shown in the fitting results of the GLS model in Table 4, the GLS–2 model resulted in better fitting results with a higher *R*^2^ value and lower values of SSE, Error^2^, and *S_e_*/*S_y_* compared to the GLS–1 model. As detailed in Table 7, the linear fitting results of the GLS–2 model were superior to that of the GLS–1 model. This was because the intercept of the fitting line of the GLS–2 model was smaller, and both the slope of fitting and *R*^2^ value were larger than that of the GLS–1 model. This suggests that the same shift factors of the dynamic modulus with the GLS–1 model can magnify the discrepancy between predicted and measured phase angle values. Consequently, the GLS–2 model offers a more precise prediction of phase angle data compared to the GLS–1 model.

For the HN model, the HN–1 model and HN–2 model represent the phase angle master curve constructed using the same WLF shift factors of the corresponding dynamic modulus and newly fitted WLF shift factors, respectively. The phase angle master curves displayed in Figure 7a,b had similar shapes but were distinct from those of the SLS and GLS models. The ranges of the master curves constructed using the HN–1 model and HN–2 model were 0 to 12° and 10°, respectively. The phase angle ranges were significantly smaller than that of LVE materials (0–90°), indicating that the HN model is unsuitable for fitting and analyzing phase angle data of the PU mixture.

Based on the analysis above, it was found that the SLS–2 and GLS–2 models exhibited higher fitting precision compared to the SLS–1 and GLS–1 models, respectively. This indicates that the models using newly fitted WLF shift factors for phase angle master curves can better predict measured phase angle data across various test temperatures. Therefore, the SLS–2 and GLS–2 models were selected for further comparison. The phase angle for LVE material is from 0 to 90°. The corresponding loading frequency for the SLS–2 model is from 10 × 10^−19.15^ to 10 × 10^20^ rad/s, while that for the GLS–2 model is from 10 × 10^−18.45^ to 10 × 10^20^ rad/s. The SLS–2 and GLS–2 model master curves are plotted together in Figure 7a. The SLS–2 and GLS–2 model master curves merged when the loading frequency exceeded 10 × 10^−5^ rad/s. At the same loading frequency, the GLS–2 model master curve was slightly higher than the SLS–2 model curve, with the difference increasing as the loading frequency decreased. As shown in Figure 8a, the predicted phase angle master curves are in good agreement with the laboratory experimental results. The fitting accuracy of the SLS–2 and GLS–2 models, as indicated in Table 3 and Table 4, suggests that both models have similar fitting accuracy. As shown in Figure 8c, the measured phase angle data points were plotted against the predicted phase angle values by SLS–2 and GLS–2 models. Table 6 and Table 7 present the linear fitting results, which show that the SLS–2 and GLS–2 models were similar and exhibit near dispersion around the line of equality. Considering the variability of phase angle measurements at different temperatures, the SLS and GLS models with newly fitted WLF shift factors can predict the phase angle by using Equations (5) and (6) with acceptably high predicting accuracy.

The phase angle data from the control group are plotted against the predicted phase angle at the same loading frequency and test temperature in Figure 8d, and the linear fitting results between predicted and measured phase angle data are listed in Table 8. Based on the linear fitting results, the SLS and GLS models could predict the phase angle data within the test temperature and loading frequency, for the *R*^2^ value was high, and the slope of the fitting linear was close to 1, which indicated that the measured phase angle data were slightly higher than the predicted data. These comparison results indicated that the phase angle data could be predicted by the phase angle master curve which was constructed by the K–K relation with relatively high precision with the test temperature and loading frequency.

The prediction of phase angle from dynamic modulus relies on the assumption of linear viscoelastic behavior in the small strain region, and materials can be considered thermorheologically simple. Additionally, the time–temperature supposition must be valid [16]. Based on the analysis above, if the dynamic modulus SLS and GLS functions for the master curves are known, the corresponding phase angle master curve SLS and GLS functions can be derived using the interconversion method in the form of the Hilbert integral transforms. Subsequently, the corresponding SLS and GLS models for phase angle, combined with the WLF function, can be utilized to fit the measured phase angle data with acceptable prediction precision. However, the parameters obtained from fitting the dynamic modulus master curve cannot be directly applied to constructing the phase angle master curve. Therefore, it is not possible to predict the phase angle of the PU mixture from its corresponding dynamic modulus data. The K–K relation did not completely apply to the PU mixture; currently, the phase angle can only be determined through a complex modulus test.

### 4.4. Core–Core and Black Space Diagrams

A lot of plots are commonly used to analyze complex modulus test results in the complex plane. For example, the core–core plot [38] plots the loss modulus against the storage modulus, or the black space plot [7] plots the phase angle versus the dynamic modulus. If the complex modulus of a thermorheologically simple HMA is measured at various temperatures and frequencies, all associated data points fall on a single curve in these diagrams [39]. In this study, the core–core and black space plots were used to analyze the complex plane behavior of the PU mixture. In this section, the fitting results of SLS–2 and GLS–2 models were used in the core–core and black space analysis.

The core–core plots of the SLS and GLS models, shown in Figure 9 displayed a single curve with one peak within a phase angle range of 0–90°. This indicated that the PU mixture exhibits thermorheologically simple properties to some extent. For the core–core plots of the SLS and GLS models shown in Figure 9, the phase angle ranged from 0 to 90°, indicating that the PU mixture was within the LVE range. When the storage modulus approached 0, the loss modulus was greater than 0, indicating that the PU mixture maintains its viscosity property at high temperatures. Therefore, the PU binder maintains its viscosity at extremely high temperatures and can be combined with the aggregate skeleton to resist high-temperature deformation under stress. As the temperature decreases, the phase angle decreases as shown in Figure 9. Both the loss modulus and storage modulus increase and the loss modulus and storage modulus reach their peak, decreasing with further temperature increases. The peak value of loss modulus is significantly lower than that of the storage modulus, indicating that the PU mixture exhibits viscosity and elastic properties that predominantly determine its dynamic behavior. The PU mixture exhibits elastic properties and only displays certain viscosity properties at extremely high temperatures. However, the PU mixture maintains its viscosity even at extremely low temperatures. The core–core plots of the SLS and GLS models display similar shapes, but there are differences in loss modulus and storage modulus values at extremely high and low temperatures. However, the peaks of the plots have nearly the same position and values. This shows both the SLS and GLS models can fit the loss modulus and storage modulus of the PU mixture. Furthermore, the loss modulus and storage modulus calculated from the dynamic modulus and phase angle can be employed to determine the LVE properties of the PU mixture at a given test temperature and loading frequency with high precision. A good indication of thermorheological simplicity is through the core–core plot, which shows the locus of the complex modulus in a complex plane as described by [40]. If all data points are close to one curve, it satisfies the requirements for thermorheological simplicity. This result confirms that the K–K relation can be used to analyze the loss modulus and storage modulus of the PU mixture.

The black space diagram can evaluate the asphalt mixture stiffness and relaxation capability through a plot of dynamic modulus as a function of phase angle. The black space diagram and core–core diagrams were used to assess the quality of rheological data and confirm the thermorheological simplicity of the asphalt mixture [41,42,43]. In this section, both measured and predicted phase angle data were employed in the black space diagram to assess the relative location of the points.

As shown in Figure 10 the black space diagrams for both the SLS and GLS models draw a single continuous curve, indicating overlapping isotherms at various temperatures. This suggests that the PU mixture possesses the thermorheologically simple property. In complex modulus tests, each phase angle data point corresponds to a single dynamic modulus value for the LVE material. The smoothness of the black space diagram indicates that the mechanical behavior, evaluated through dynamic modulus and phase angle, can be determined across a range of temperatures and loading frequencies. This is consistent with time–temperature equivalence. As shown in Figure 10, the black space diagrams for both SLS and GLS models decreased with an increasing phase angle, and they do not exhibit an inflection point during this change, unlike the black space diagram of HMA. This suggests that, in comparison to HMA, the PU mixture possesses reduced rheological properties and is less impacted by the temperature, further supporting its thermorheologically simple property. The black space diagrams for the SLS and GLS models are plotted together in Figure 10, and the black space diagrams demonstrate nearly identical shapes and values for SLS and GLS models. This suggests that the model type does not significantly impact the rheological property of the PU mixture.

## 5. Conclusions

In this study, the complex modulus test was performed on a PU mixture. The SLS, GLS, and HN models were used to fit the dynamic modulus test results. The corresponding SLS, GLS, and HN models for the phase angle were calculated from the dynamic modulus master curve models using K–K relations. These models were then adopted to fit the phase angle test results, utilizing the same WLF shift factors from the dynamic modulus master curve and a newly fitted WLF shift factor. The core–core and black space diagrams were employed to analyze the thermorheologically simple properties and rheology properties of the PU mixture. Based on the analysis of the complex modulus test results, the following conclusions can be drawn.

(1) The dynamic modulus and phase angle gradually change with increasing test temperatures and loading frequencies. The phase angle does not show peaks, and the PU mixture exhibits more obvious thermorheologically simple properties.

(2) The PU mixture demonstrates LVE properties with a low strain level (<100 με).

(3) The referenced temperatures only shift the master curve to the right or left and do not influence its shape. The SLS, GLS, and HN models produced similar WLF shift factors.

(4) The SLS and GLS models produced similar fitting results for the dynamic modulus which were slightly better than that of the HN model. All three models (SLS, GLS, and HN) had good agreement with the experimentally measured dynamic modulus data.

(5) The predicted phase angle range fitted by the HN model for the measured phase angle data was too small; thus, the HN model was not appropriate for the PU mixture in terms of phase angle.

(6) The SLS and GLS models with newly fitted WLF shift factors exhibited better prediction accuracy compared to those using the same WLF shift factors from the dynamic modulus master curve.

(7) The selected SLS and GLS models for the phase angle showed similar shapes, shift factors, and predicting accuracy.

(8) The phase angle of the PU mixture can be calculated from the corresponding experimentally measured dynamic modulus data based on the K–K relation within the test temperature and loading frequency.

(9) The core–core and black space diagrams both depicted a single, continuous, smooth curve. However, the latter diagram did not show inflection points. The rheological property of the PU mixture was significantly smaller than that of HMA. Moreover, the PU mixture shows obvious thermorheologically simple properties.

As discussed previously, when combined with the WLF shift factor, the SLS and GLS models can accurately predict the experimental dynamic modulus. By utilizing the K–K relations, the corresponding SLS and GLS models derived from the dynamic modulus master curve equation can establish the master curves for phase angle, exhibiting a strong agreement with the experimentally measured phase angle data. Nevertheless, the WLF shift factors for the SLS and GLS models of the phase angle must be newly fitted to achieve high fitting accuracy. As a result, the phase angle master curve equation can be calculated from the corresponding dynamic modulus master curve equation by utilizing the K–K relations, and the calculated equation can fit the experimentally measured phase angle data of the PU mixture with high accuracy with the test temperature and loading frequency. However, the phase angle data of the PU mixture at extremely high or low test temperatures cannot be calculated from the corresponding dynamic modulus. The K–K relations only partially apply to the PU mixture. This finding confirmed that the rheological property of the PU mixture was different from that of HMA. At present, there is no special dynamic modulus master curve equation for PU mixtures, and all dynamic modulus master curve equations were introduced from HMA. PU mixture is a new kind of pavement mixture that needs more studying to develop its own dynamic modulus master curve equation.

Currently, it is impossible to obtain the phase angle data of the PU binder at extremely high temperatures in laboratory conditions. Therefore, the actual phase angle range of the PU mixture or binder remains unknown. In the future, it will be beneficial to study the phase angle of the PU binder at extremely high temperatures through dynamic mechanical analysis. The proper equations that can accurately fit the dynamic modulus and calculate the phase angle data from the dynamic modulus of the PU mixture should be developed.

## Figures and Tables

**Figure 1 materials-17-02909-f001:**
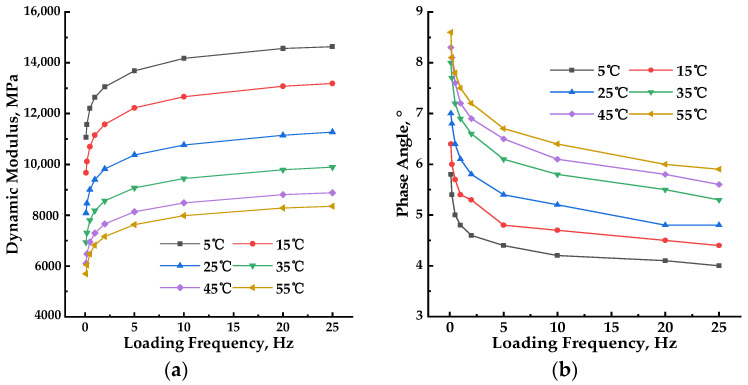
The dynamic modulus test results of the PU mixture. (**a**) Dynamic modulus results under different test temperatures; (**b**) phase angle results under different test temperatures.

**Figure 2 materials-17-02909-f002:**
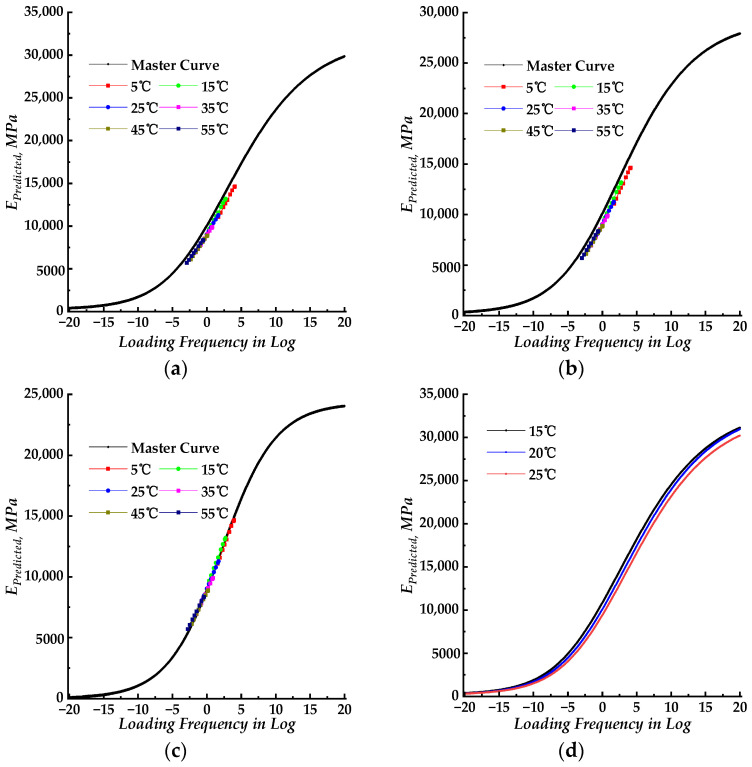
The dynamic modulus master curve for different models at the same reference temperature. (**a**) Master curve for the SLS model with shifted measured dynamic modulus data; (**b**) master curve for the GLS model with shifted measured dynamic modulus data; (**c**) master curve for the HN model with shifted measured dynamic modulus data; (**d**) master curves for the SLS model at different reference temperatures.

**Figure 3 materials-17-02909-f003:**
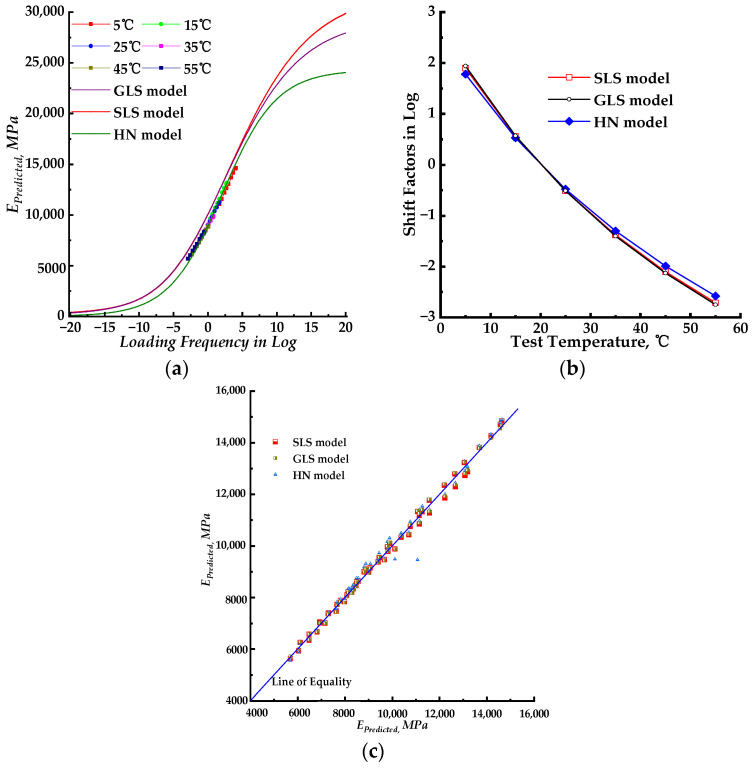
Comparing the fitting results for the three mathematical models. (**a**) Dynamic modulus master curves with shifted measured dynamic modulus data; (**b**) shifted factors for three models; (**c**) predicted dynamic modulus versus measured dynamic modulus.

**Figure 4 materials-17-02909-f004:**
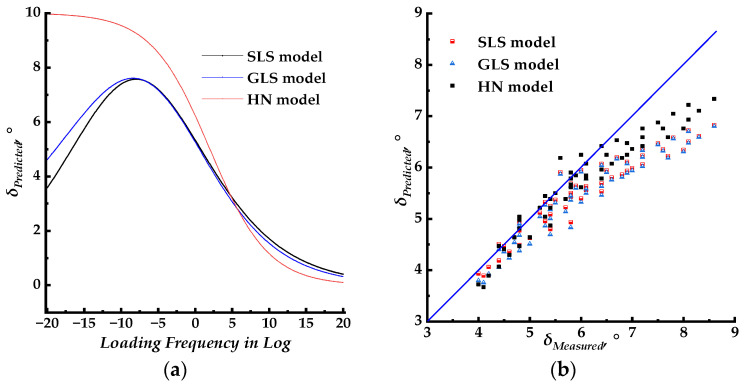
The phase angle master curves constructed using the fitted parameters of the dynamic modulus master curve model. (**a**) Phase angle master curves with the fitting parameters of dynamic modulus master curves; (**b**) predicted phase angle versus measured phase angle.

**Figure 5 materials-17-02909-f005:**
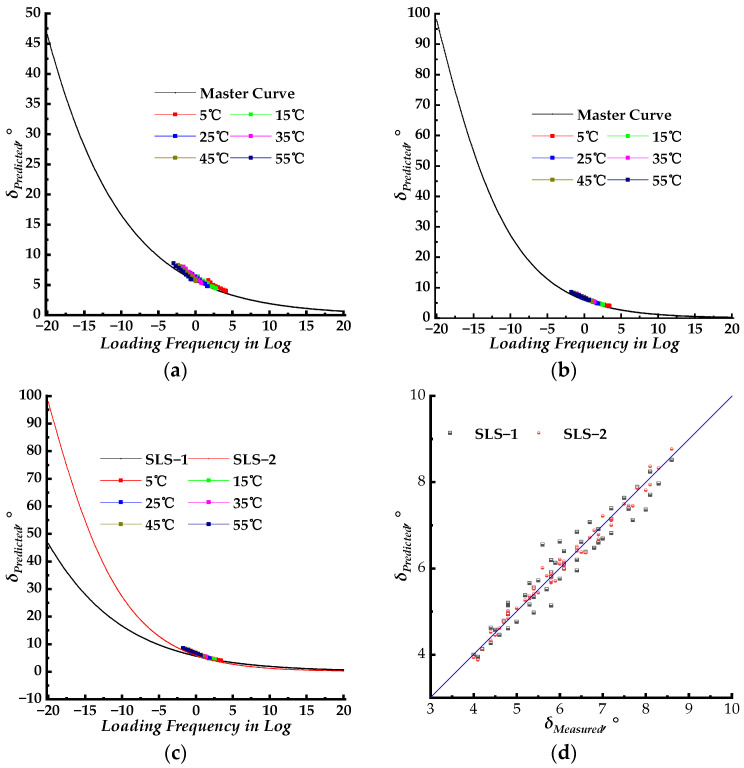
The phase angle master curves for the SLS model. (**a**) Master curve with identical WLF shift factors of the dynamic modulus (SLS–1 model); (**b**) master curve with newly fitted shift factors (SLS–2 model); (**c**) master curves with shifted measured phase data; (**d**) predicted phase angle versus measured phase angle.

**Figure 6 materials-17-02909-f006:**
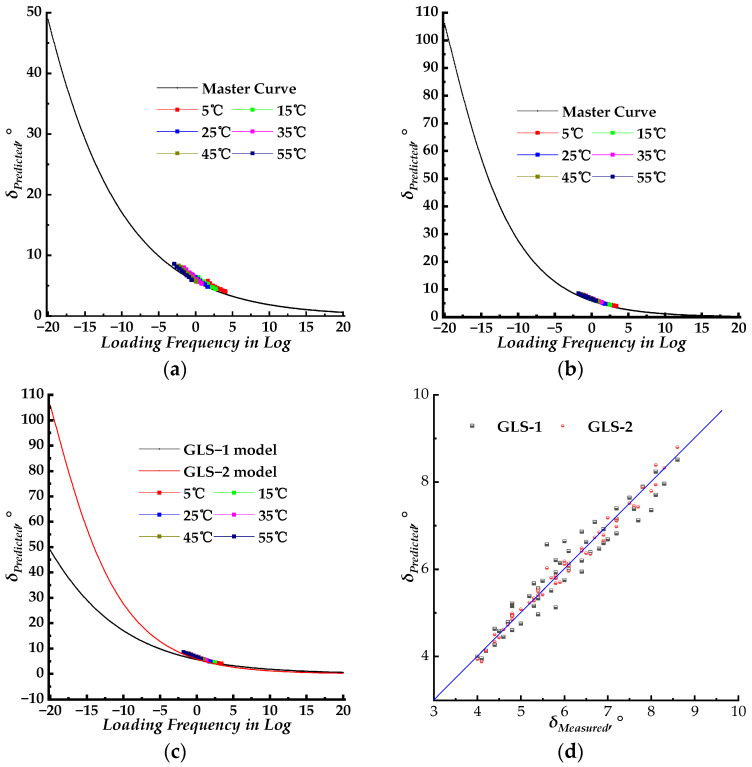
The phase angle master curves for the GLS model. (**a**) Master curve with identical WLF shift factors of the dynamic modulus (GLS–1 model); (**b**) master curve with newly shift fitted factors (GLS–2 model); (**c**) master curves with shifted measured phase data; (**d**) predicted phase angle versus measured phase angle.

**Figure 7 materials-17-02909-f007:**
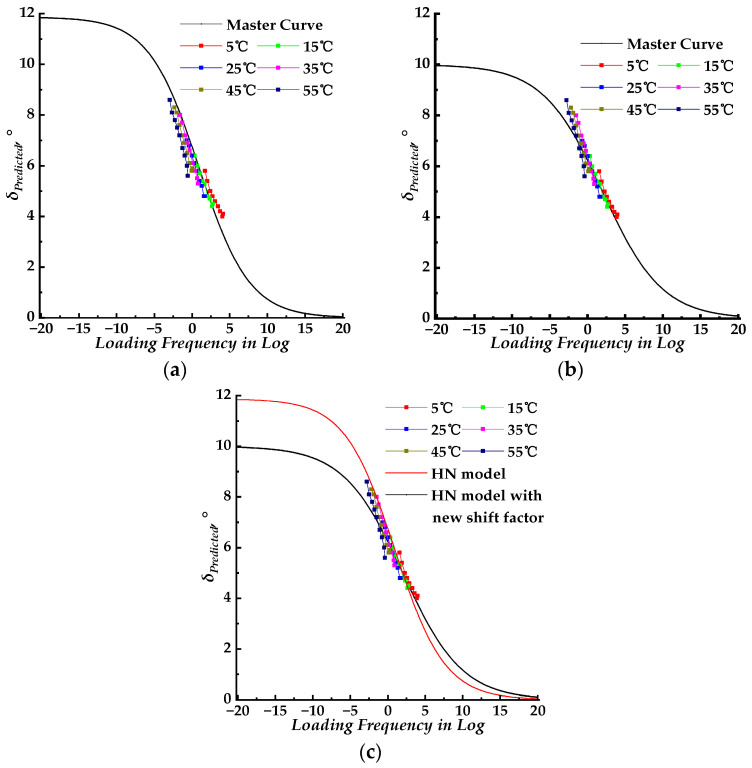
The phase angle master curves for the HN model. (**a**) Master curve with identical WLF shift factors of the dynamic modulus (HN–1 model); (**b**) master curve with newly shift fitted factors (HN–2 model); (**c**) master curves with shifted measured phase data.

**Figure 8 materials-17-02909-f008:**
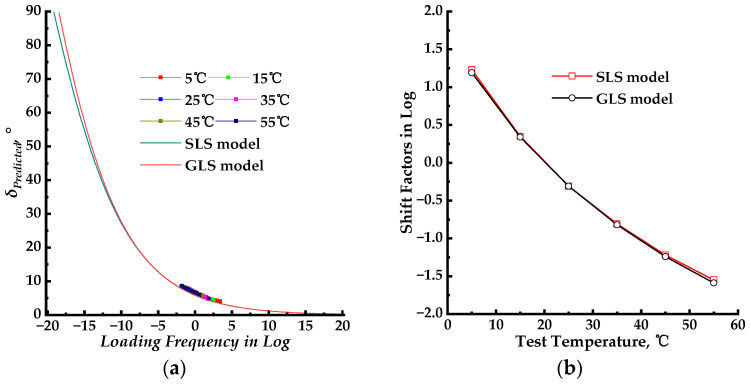
Comparing the fitting results for SLS and GLS models. (**a**) Phase angle master curves with shifted measured phase angle data; (**b**) shifted factors for SLS and GLS models; (**c**) predicted phase angle of SLS and GLS models versus measured phase angle; (**d**) predicted phase angle of SLS and GLS models versus measured phase angle from the control group.

**Figure 9 materials-17-02909-f009:**
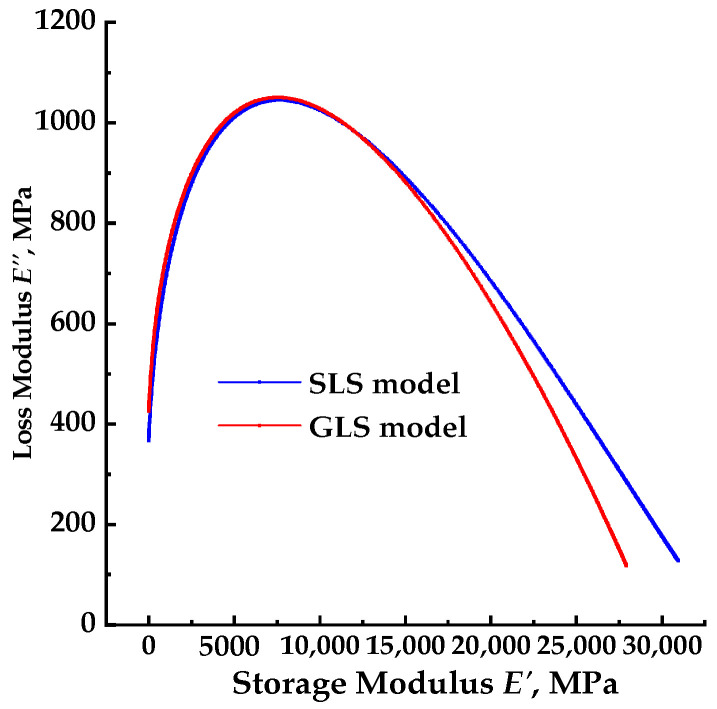
The core–core diagrams for the SLS and GLS models.

**Figure 10 materials-17-02909-f010:**
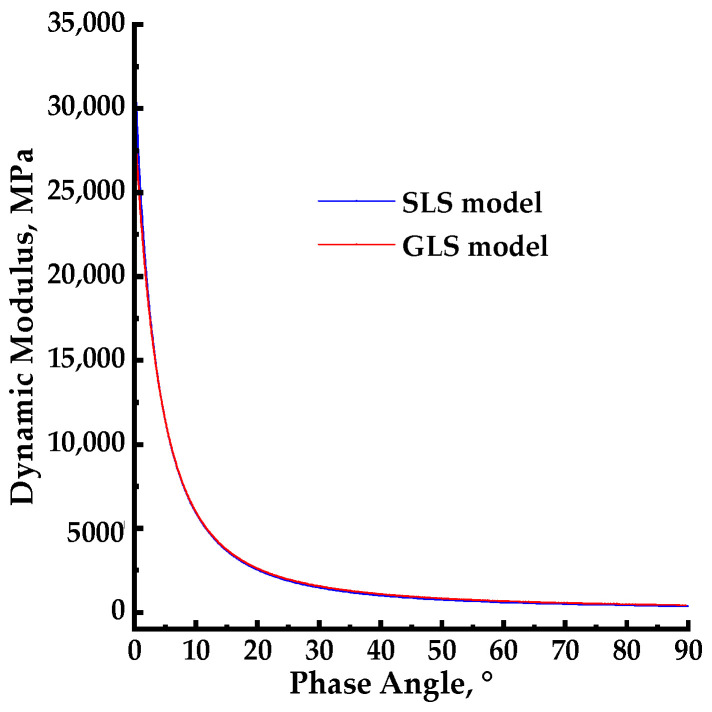
The black space diagrams for the SLS and GLS models.

**Table 1 materials-17-02909-t001:** The aggregate gradation for the PU mixture.

Sieve/mm	0.075	0.15	0.3	0.6	1.18	2.36	4.75	9.5	13.2	16
Passing/%	5.2	6.3	8.1	11.7	19	29.8	38.8	76.7	98.1	100

**Table 2 materials-17-02909-t002:** The fitting accuracy results for the SLS, GLS, and HN models.

Model	*R* ^2^	SSE	Error^2^	*Se*/*Sy*
SLS	1	1.48 × 10^−2^	3.00 × 10^−3^	4.55 × 10^−4^
GLS	1	1.50 × 10^−2^	3.00 × 10^−3^	4.44 × 10^−4^
HN	1	4.70 × 10^−2^	1.00 × 10^−3^	4.90 × 10^−4^

**Table 3 materials-17-02909-t003:** The fitting accuracy results for the SLS–1 and SLS–2 models.

Model	*R* ^2^	SSE	Error^2^	*Se*/*Sy*
SLS–1	0.9821	0.136	5.063	9.79 × 10^−2^
SLS–2	0.9996	0.023	0.853	1.47 × 10^−2^

**Table 4 materials-17-02909-t004:** The fitting accuracy results for the GLS–1 and GLS–2 models.

Model	*R* ^2^	SSE	Error^2^	*Se*/*Sy*
GLS–1	0.9806	0.143	5.288	1.02 × 10^−1^
GLS–2	0.9995	0.022	0.857	1.66 × 10^−2^

**Table 5 materials-17-02909-t005:** The linear fitting results for predicted and measured dynamic modulus data.

Model	Equation	*R* ^2^
SLS	Y = 41.8 + 0.9955X	0.9948
GLS	Y = 48.1 + 0.9956X	0.9948
HN	Y = 75.2 + 0.9964X	0.9839

**Table 6 materials-17-02909-t006:** The linear fitting results between predicted and measured dynamic modulus data for the SLS model.

Model	Equation	*R* ^2^
SLS–1	Y = 0.4178 + 0.9278X	0.9295
SLS–2	Y = 0.1088 + 0.9827X	0.9859

**Table 7 materials-17-02909-t007:** The linear fitting results between predicted and measured dynamic modulus data for the GLS model.

Model	Equation	*R* ^2^
GLS–1	Y = 0.3882 + 0.9335X	0.9315
GLS–2	Y = 0.0831 + 0.9859X	0.9857

**Table 8 materials-17-02909-t008:** The linear fitting results between predicted and measured dynamic modulus data from the control group.

Model	Equation	*R* ^2^
SLS–2	Y = −0.2536 + 1.0062X	0.9721
GLS–2	Y = −0.2858 + 1.0104X	0.9736

## Data Availability

Data are contained within the article. Please refer to the complete guideline at https://doi.org/10.3390/coatings13071143.

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
