# Peer review of "Study on the Application of Kramers–Kronig Relation for Polyurethane Mixture"

_materials, 2024, doi:10.3390/ma17122909_

Round 1
Reviewer 1 Report (New Reviewer)
Comments and Suggestions for Authors The current manuscript is interesting, but it requires revision prior to its acceptance, please see and address the points listed below. Towards the end of the introduction the authors should concisely but clearly state what they do, how they do it (methods, approaches, models, etc.), and what exactly do they get as the results. A sectioned plan of the entire paper is to be presented here. First reference to an equation should appear in the text after the equation itself (not before it); sentences not following this rule should be reformulated. All figures should be of a unified size (8 cm in width for single-column plots), of the same style (regarding the frames, ticks, geometric proportions, axis-caption fonts (size and style), etc.). The quantities presented in the plots should (at best) have their full name, mathematical notation, and physical units being presented along each axis, all separated by commas, in order to facilitate the perception by the reader. Axis captions of the plots are often way too small. The font size for the axis captions should be the same or even larger than the font size of the standard text in the finally formatted version of the paper. The caption of each figure should be self-explanatory and also understandable without reading a respective part of the text.The discussion is long but badly structured. The conclusions should be presented in a simple---possibly even numbered---manner. What was the initial supposition or an expectation regarding the general principles of physical functioning of a given system and what do the results of the analysis of collected experimental data or the evidence from computer simulations or theory indicate? This message should be delivered crystal-clear; it should be understandable not only for experts, but also for pedestrians/laypersons. If many (10+) acronyms are in use (such as in a long review, for instance), the list of all abbreviations as well as of the nomenclature is to be provided prior to the bibliography for readers' convenience. Interesting, the viscoelastic properties of asphalt studied by the authors occur also e.g. in the cytoplasm of biological cells, where a variety of conditions regulate the response. A number of recent studies examined e.g. how the distributed viscosity can control the overall elastic response of such viscoelastic media, please consult and mention in the revised version Ref. [https://doi.org/10.1039/D2SM00952H]. What would be authors' expectation regarding the overall response of a complicated multi-component mixture of materials in asphalt, for instance, where---similarly to cells of glycerol mixtures---each of the components has its own and specifically varying viscoelastic portrait. This is an important issue to touch upon. Do not use a sequence of separated nouns as adjectives: such constructions have unclear meaning and are hard to read. Sequences of three and more nouns must in such cases be hyphenated to avoid ambiguity and to facilitate reading. The same ambiguity emerges e.g. when using ill-defined multiple-division math operations, such as a/b/c, a/b/c/d, etc.
Author Response
I appreciate your suggestions and valuable advice. Based on the suggestions, I modified the manuscripts as follows.
Point 1: Towards the end of the introduction the authors should concisely but clearly state what they do, how they do it (methods, approaches, models, etc.), and what exactly do they get as the results. A sectioned plan of the entire paper is to be presented here.
Response 1: More information about the PU mixture and objectives of this paper were added to the introduction in the new manuscript. Methods, approaches, models, and section plan were shown in the end part of the introduction in lines of 97-104. The findings of this paper were scattered, and I think it was not proper to present the findings in the section.
Point 2: First reference to an equation should appear in the text after the equation itself (not before it); sentences not following this rule should be reformulated.
Response 2: In the new manuscript, all first sentences related to equations were reformulated and removed to follow equations.
Point 3: All figures should be of a unified size (8 cm in width for single-column plots), of the same style (regarding the frames, ticks, geometric proportions, axis-caption fonts (size and style), etc.). The quantities presented in the plots should (at best) have their full name, mathematical notation, and physical units being presented along each axis, all separated by commas, in order to facilitate the perception by the reader. Axis captions of the plots are often way too small. The font size for the axis captions should be the same or even larger than the font size of the standard text in the finally formatted version of the paper. The caption of each figure should be self-explanatory and also understandable without reading a respective part of the text.
Response 3: The required size of a single-column plot should be 6 cm in the template. All plots followed the same style (regarding the frames, ticks, geometric proportions, axis-caption fonts (size and style), etc.). The quantities presented in the plots were rewritten in the new manuscript. Axis captions of the plots were enlarged in new manuscript. The template required detailed clarification about each plot in the title, therefore, the caption of each figure was shortened for clear representation.
Point 4: The discussion is long but badly structured. The conclusions should be presented in a simple---possibly even numbered---manner. What was the initial supposition or an expectation regarding the general principles of physical functioning of a given system and what do the results of the analysis of collected experimental data or the evidence from computer simulations or theory indicate? This message should be delivered crystal-clear; it should be understandable not only for experts, but also for pedestrians/laypersons. If many (10+) acronyms are in use (such as in a long review, for instance), the list of all abbreviations as well as of the nomenclature is to be provided prior to the bibliography for readers' convenience.
Response 4: The discussion part corresponds to the part of the result. The conclusion part was partly rewritten in the new manuscript. All the acronyms were listed in the manuscript, but the template does not provide a list of acronyms part.
Point 5: Interesting, the viscoelastic properties of asphalt studied by the authors occur also e.g. in the cytoplasm of biological cells, where a variety of conditions regulate the response. A number of recent studies examined e.g. how the distributed viscosity can control the overall elastic response of such viscoelastic media, please consult and mention in the revised version Ref. [https://doi.org/10.1039/D2SM00952H].
Response 5: Thank you for providing a new aspect about viscoelastic properties, the PU binder would be in a solid phase state and its phase state can not be changed with the rising of temperature which is different from asphalt or cytoplasm of biological cells.
Point 6: What would be authors' expectation regarding the overall response of a complicated multi-component mixture of materials in asphalt, for instance, where---similarly to cells of glycerol mixtures---each of the components has its own and specifically varying viscoelastic portrait. This is an important issue to touch upon.
Response 6: In this paper, asphalt is not the studying material, the PU is the studying material. PU is an industrial compound that is different from asphalt, for asphalt is a mixture of different components. Therefore, this point was not involved in this study.
Point 7: Do not use a sequence of separated nouns as adjectives: such constructions have unclear meaning and are hard to read. Sequences of three and more nouns must in such cases be hyphenated to avoid ambiguity and to facilitate reading. The same ambiguity emerges e.g. when using ill-defined multiple-division math operations, such as a/b/c, a/b/c/d, etc.
Response 7: There are some sequences of three nouns, e.g., thermo−rheological simple properties, and time−temperature superposition principle. These expressions were the traditional expression, if all the nouns were combined with hyphenated, it may provide confusion for scholars.
Thank you again for your hard, meticulous work!
Best regards!

Reviewer 2 Report (New Reviewer)
Comments and Suggestions for Authors
The paper is interesting. It has the potential to contribute to the knowledge. The highlight of the study is the combination of the experimental investigation coupled with mathematical modelling to verify if K-K relations apply to the PU mixture.
However, the paper needs to be improved from the methodological and study’s implementation point of view. In this context, first, the authors need to clarify the objective of the study. In other words, they need to make it clear what they mean by verifying the application of K-K relation to the PU mixture. Furthermore, they should explain in what context the relationship is intended to be verified. Before that, they must also explain the fundamentals and implications of K-K relations and PU mixture and related models used.
Methodologically, since experimental investigation is a major part of the study, a detailed investigation process may be provided. For example, what experimental was set up, the procedure to collect the data, how many samples were used, how samples were created, adequacy of sample size, and so on. Also, it is not clear why a specimen size of 150 mm in height and 100 mm in diameter was used. Does this have any impact on the results?
Furthermore, the authors have used several mathematical models. However, it is unclear why they are relevant and whether they are used concurrently or in isolation. Also, it is not clear how the results were obtained by using these models. If any software or algorithm is used, then it should be mentioned.
From the results point of view, it seems the authors have not clearly articulated what is the outcome of this investigation. In other words, what is the relationship between K-K relations and PU mixture under different conditions, if different conditions were used? The authors also need to discuss the practical and theoretical implications of the findings.
Comments on the Quality of English LanguageModerate language editing is necessary to make the articulation lucid.
Author Response
I appreciate your suggestions and valuable advice. Based on the suggestions, I modified the manuscripts as follows.
Point 1: However, the paper needs to be improved from the methodological and study’s implementation point of view. In this context, first, the authors need to clarify the objective of the study. In other words, they need to make it clear what they mean by verifying the application of K-K relation to the PU mixture. Furthermore, they should explain in what context the relationship is intended to be verified. Before that, they must also explain the fundamentals and implications of K-K relations and PU mixture and related models used.
Response 1: The objective of the article was revised, the main objective of this paper is to verify when dynamic modulus data of PU mixture are available, and whether phase angle data can be obtained from corresponding dynamic modulus data in terms of the K-K relations. The relationship should be used in test results of indirect tensile test or fatigue test in which only dynamic modulus data can be obtained. The fundamentals and implications of K-K relations and PU mixture and related models were listed in the new manuscript in lines 66-68, 87-90, and equations (1)-(4).
Point 2: Methodologically, since experimental investigation is a major part of the study, a detailed investigation process may be provided. For example, what experimental was set up, the procedure to collect the data, how many samples were used, how samples were created, adequacy of sample size, and so on. Also, it is not clear why a specimen size of 150 mm in height and 100 mm in diameter was used. Does this have any impact on the results?
Response 2: More information about samples was added in the new manuscript. Eight specimens were performed for the dynamic modulus test, and each four duplicates was divided into one group, the first group was the control group, and the second group was the verifying group. The average values of four duplicates are enough for test results discussion according to the specification. The dynamic modulus test condition was shown in the manuscript (e.g., six temperatures (5℃, 15℃, 25℃, 35℃, 45℃, and 55℃,) and nine loading frequencies (25, 20, 10, 5, 2, 1, 0.5, 0.2, and 0.1 Hz)). Dynamic modulus tests were performed at different test temperatures and loading frequencies, and then dynamic modulus and phase angle data can be obtained at different test temperatures and loading frequencies. The dynamic modulus test followed the AASHTO: TP−79 (2010) specification which required the standard specimen size should be 150 mm in height and 100 mm in diameter.
Point 3: Furthermore, the authors have used several mathematical models. However, it is unclear why they are relevant and whether they are used concurrently or in isolation. Also, it is not clear how the results were obtained by using these models. If any software or algorithm is used, then it should be mentioned.
Response 3: I am sorry about the confusion. Equations (5)-(7) were individual equations, which represent the SLS, GLS, and HN models, respectively. Equations (9)-(11) were individual equations obtained from equations (5)-(7) by equation (8), respectively. Equations (12)-(13) were used to replace the f(r) or ω parameter in equations (5)-(7) which related to temperature. A nonlinear regression procedure as shown in (14)-(15) was performed to minimize the difference between measured data and predicted data by equations (5)-(7) and (9)-(11). The Excel solver was used in the nonlinear regression procedure, and this sentence was added to the new manuscript. Equations (16)-(20) were used as the main indexes for the Excel solver regression procedure.
Point 4: From the results point of view, it seems the authors have not clearly articulated what is the outcome of this investigation. In other words, what is the relationship between K-K relations and PU mixture under different conditions, if different conditions were used? The authors also need to discuss the practical and theoretical implications of the findings.
Response 4: The concept of K-K relations was introduced into the hot mixed asphalt mixture to predict phase angle data from corresponding dynamic modulus when phase angle data was lost or not measured at the same time. In this paper, the concept of K-K relations was adopted to predict phase angle data from the corresponding dynamic modulus for PU mixture and to compare predicted phase angle data with measured data for verifying the application of K-K relations. The test result confirmed that the predicted phase angle in terms of K-K relations was not in line with measured phase angle data, therefore, the concept of K-K relations does not apply to PU mixture. This condition was the only condition for K-K relations and PU mixture at present. More practical and theoretical implications of these findings were added to the new manuscript.
Point 5: Moderate language editing is necessary to make the articulation lucid.
Response 5: I have rechecked the manuscript, and many sentences were rewritten for lucid expression.
Thank you again for your hard, meticulous work!
Best regards!

Reviewer 3 Report (New Reviewer)
Comments and Suggestions for Authors
Review manuscript ID: materials-2994472
Type of manuscript: Article
Title: Study on the Application of Kramers−Kronig Relation for the Polyurethane Mixture
Authors: Haisheng Zhao, Quanjun Shen, Peiyu Zhang, Zhen Li, Shiping Cui, Lin Wang, Wensheng Zhang, Chunhua Su, Shijie Ma*
Journal: Materials
Section: Polymeric Materials
Publisher: MDPI, Basel, Switzerland
Assistant Editor: Mr. Octavian Sergiu Barbos
Date: 22 May 2024
This study examined whether the Kramers-Kronig (K-K) relation and thermo-rheological simple properties apply to Polyurethane (PU) mixtures, a high-performance material differing from hot-mixed asphalt mixtures. Results showed that PU mixtures exhibit thermo-rheological simple properties under test conditions. The following provides some remarks.
On page 2, the sentence should precede before the equations, not the other way around. Please also explain why the approximate symbols were used instead of equal signs. Please confirm whether the meaning of the prime symbol denotes the first derivative. The same remark for the operator d/d[something]. Does ln denote the natural logarithm? Note that in other expressions, the authors use log instead of ln. I think this is confusing to many readers. By the way, neither ln nor log should be written in italics mode.
Wordy phrases: were applicable --> applied
It was not clear what variable the function alpha depended on.
There was no integral when discussing the Hilbert integral transform.
Why is an absolute value necessary in (8)? Shouldn't the modulus be nonnegative?
How do we know that lambda will never vanish?
The use of excessive dot product can be easily mistaken as cross product.
Please confirm whether an asterisk denotes the complex conjugate.
There are unnecessary vertical lines in the figure. Please remove them. The figure labels should also be enlarged.
Remove the symbol times.
No space before the degree symbol.
Comments on the Quality of English Language
Clarity issues.
Author Response
I appreciate your suggestions and valuable advice. Based on the suggestions, I modified the manuscripts as follows.
Point 1: On page 2, the sentence should precede before the equations, not the other way around. Please also explain why the approximate symbols were used instead of equal signs. Please confirm whether the meaning of the prime symbol denotes the first derivative. The same remark for the operator d/d[something]. Does ln denote the natural logarithm? Note that in other expressions, the authors use log instead of ln. I think this is confusing to many readers. By the way, neither ln nor log should be written in italics mode.
Response 1: The sentence was moved before the equations.
The Kramers-Kronig (K-K) relation represented in equations(1)-(4) were the highlight results of reference 18 “Generalization of Kramers-Kronig transforms and some approximations of relations between viscoelastic quantities”. In reference 18, the equal signs of “=” were replaced with congruence signs of “≅”, and the approximate symbols were replaced with congruence signs in the new manuscript.
The prime symbol and operator d/d[something] denote the first derivative.
The mistakes about the symbol of “ln” were corrected and replaced with “log”, and all the signs of “log” in italics mode were corrected in the new manuscript.
Point 2: Wordy phrases: were applicable --> applied?
Response 2: The phrases “were applicable” were replaced with “applied” in the new manuscript.
Point 3: It was not clear what variable the function alpha depended on.
Response 3: In equation (13) about the function alpha, the variable was the temperature “T”. The reference was set at 20℃, the equation (13) can be used to calculate the shift factor value of α(T) at different test temperatures of “T”.
Point 4: There was no integral when discussing the Hilbert integral transform.
Response 4: The integral process of equations (5)-(7) based on the Hilbert integral transform was not discussed in this paper, for the integral process was complicated and not the main focus of this paper. Therefore, the integral process was not involved in this paper, and the integral results were shown in the equations (9)-(11).
Point 5: Why is an absolute value necessary in (8)? Shouldn't the modulus be nonnegative?
Response 5: I am sorry about the confusion. In this paper, the sign of “E*” represents the complex dynamic modulus, as shown in this equation E*=E’+i*E’’, therefore, the “E*” was not nonnegative, it should be used as an absolute value in equation (8).
Point 6: How do we know that lambda will never vanish?
Response 6: During the fitting process, there would be a lot of additional indexes used for obtaining the optimum results, the parameter of lambda can be set not equal to zero as the additional index. Then, the lambda would not vanish during the fitting process.
Point 7: The use of excessive dot product can be easily mistaken as cross product.
Response 7: I am sorry about this confusion. The main integral process was introduced from reference 18 and was verified by many scholars, i.g., reference 34. Then, I did not pay much attention to the integral process.
Point 8: Please confirm whether an asterisk denotes the complex conjugate.
Response 8: I confirmed that the asterisk E* as shown denotes the complex conjugate as shown in response 5.
Point 9: There are unnecessary vertical lines in the figure. Please remove them. The figure labels should also be enlarged.
Response 9: All the suggestions were accepted, and the unnecessary vertical lines were removed. Figures were replotted in the new manuscript.
Point 10: Remove the symbol times.
Response 10: The symbol times were removed.
Point 11: No space before the degree symbol.
Response 11: Spaces were added before the degree symbol.
Thank you again for your hard, meticulous work!
Best regards!

Round 2
Reviewer 2 Report (New Reviewer)
Comments and Suggestions for Authors
The paper is improved. However, articulation in terms of the presentation of the methodological approach and results is necessary to enhance clarity.
Comments on the Quality of English LanguageModerate English language editing is necessary.
Author Response
I appreciate your suggestions and valuable advice. Based on the suggestions, I modified the manuscripts as follows.
Point 1: The paper is improved. However, articulation in terms of the presentation of the methodological approach and results is necessary to enhance clarity.
Response 1: Part of the methods and results discussion was improved to provide more information for readers.
Point 2: Moderate English language editing is necessary.
Response 2: Grammar checking and correcting were taken for the new manuscript.
Thank you again for your hard, meticulous work!
Best regards!
Reviewer 3 Report (New Reviewer)
Comments and Suggestions for Authors
Second review manuscript ID: materials-2994472
Type of manuscript: Article
Title: Study on the Application of Kramers−Kronig Relation for the Polyurethane Mixture
Authors: Haisheng Zhao, Quanjun Shen, Peiyu Zhang, Zhen Li, Shiping Cui, Lin Wang, Wensheng Zhang, Chunhua Su, Shijie Ma*
Journal: Materials
Section: Polymeric Materials
Publisher: MDPI, Basel, Switzerland
Assistant Editor: Mr. Octavian Sergiu Barbos
Date: 31 May 2024
Thank you for revising the manuscript.
The notations are not explained. For example, what do G, G_d, and delta(omega) mean?
Sometimes there are thick primes, and other times thin double primes. Many readers wonder about this. I strongly encourage the authors to typeset their manuscript in LaTeX for a beautiful mathematical display.
The word error in the equation should not be in italics.
The vertical line on page 6 has not been removed. (The previous remark has been ignored.)
Figure labels are too small and should be enlarged.
The symbol "times" has not been removed. (The previous remark has been ignored.)
soso
Author Response
I appreciate your suggestions and valuable advice. Based on the suggestions, I modified the manuscripts as follows.
Point 1: The notations are not explained. For example, what do G, G_d, and delta(omega) mean?
Response 1: The explanation about these parameters were added in lines of 70-72 in the new manuscript.
Point 2: Sometimes there are thick primes, and other times thin double primes. Many readers wonder about this. I strongly encourage the authors to typeset their manuscript in LaTeX for a beautiful mathematical display.
Response 2: Thank you for your suggestion, I will learn about the LaTeX for editing the manuscript. For the equations in this paper, I will contact the editors for modifying.
Point 3: The word error in the equation should not be in italics.
Response 3: The word error was corrected in the new manuscript.
Point 4: The vertical line on page 6 has not been removed. (The previous remark has been ignored.)
Response 4: The vertical lines on page 6 were the mark about modifying, the vertical lines were deleted in the new manuscript.
Point 5: Figure labels are too small and should be enlarged.
Response 5: All the figure labels were enlarged in the new manuscript from 16 to 20 in bold form.
Point 6: The symbol "times" has not been removed. (The previous remark has been ignored.)
Response 6: I didn’t fully understand the previous remark, the symbol “times” in Tables 5, 6, 7 and 8 were removed in the new manuscript.
Thank you again for your hard, meticulous work!
Best regards!

This manuscript is a resubmission of an earlier submission. The following is a list of the peer review reports and author responses from that submission.
Round 1
Reviewer 1 Report
Comments and Suggestions for Authors
1. Introduction
1.1) L. 52: * studies
1.2) I think that the construction of the last paragraph should be revised by the authors. I suggest replacing the various objectives with a single central question that would be the heart of the work. In other words, what is the problem, in fact, that the authors intend to solve?
2. Materials and Methods
OK.
3. Results
I know that the journal admits separating the results and discussion sections. However, "instruction for authors" states that these sections can be combined. Due to the enormous volume of results, from my point of view, it would be better for the authors to present the results and then discuss them. This would make reading more enjoyable.
4. Discussion
Eliminate the space between the numeric value and the % and ° signs. For example, L. 280: 43%, etc. and L. 416: 9°, etc.
5. Conclusions
The authors could present the conclusions in a more synthetic way.
Author Response
I appreciate your suggestions and valuable advice. Based on the suggestions, I modified the manuscripts as follows.
Point 1: 1.1) L. 52: * studies
Response 1: Thank you for the suggestion. I have corrected the mistake.
Point 2: 1.2) I think that the construction of the last paragraph should be revised by the authors. I suggest replacing the various objectives with a single central question that would be the heart of the work. In other words, what is the problem, in fact, that the authors intend to solve?
Response 2: The objective of the article was revised, and the former various objectives were changed into sections or steps to fulfill the objective.
Point 3: 3. Results
I know that the journal admits separating the results and discussion sections. However, "instruction for authors" states that these sections can be combined. Due to the enormous volume of results, from my point of view, it would be better for the authors to present the results and then discuss them. This would make reading more enjoyable.
Response 3: It would provide an enjoyable reading experience if the results and discussion section were combined. I will contact the editor for revision.
Point 4: Eliminate the space between the numeric value and the % and ° signs. For example, L. 280: 43%, etc. and L. 416: 9°, etc.
Response 4: All the spaces were deleted in the new manuscript.
Point 5: The authors could present the conclusions in a more synthetic way.
Response 5: Various sections or steps were carried out to fulfill the objective of this paper, and then every section or step had individual discussion results and conclusion. I want to provide more information about the properties of the PU mixture for the reader to avoid confusion.
Thank you again for your hard, meticulous work!
Best regards!

Reviewer 2 Report
Comments and Suggestions for Authors
This was an innovative and interesting paper.
The author should provide more information on the PU mixture and its properties and reference any previous work done with the PU mixture. It may be good to compare these mixtures with an asphalt mixture made with conventional binders.
Author Response
I appreciate your suggestions and valuable advice. Based on the suggestions, I modified the manuscripts as follows.
Point 1: The author should provide more information on the PU mixture and its properties and reference any previous work done with the PU mixture. It may be good to compare these mixtures with an asphalt mixture made with conventional binders.
Response 1: I am sorry about the confusion in this paper. The information on the PU mixture and its properties and the comparison between the PU mixture and mixtures with an asphalt binder were all discussed in the reference ”https://doi.org/10.3390/coatings13071143”. If the information discussed in formerly published articles were contented in this paper, the journal would warn for repetition rate.
Thank you again for your hard, meticulous work!
Best regards!

Reviewer 3 Report
Comments and Suggestions for Authors
The manuscript experimentally analyses the applicability of the master curve procedure in linear viscoelasticity for the dynamic modulus and phase angle for a polyurethance mixture.
From a methodical point of view, I have a serious concern about the manuscript: If one performs experiments both to parametrize a model, and to test its predictions, the experimental data must be split into two sets: one for the determination of the model parameters, and one for testing the predictions of the parametrized model. In the manuscript, if I understand correctly, the same experimental data is used for fitting the parameters and to compare the model predictions with measured results (except only for the comparison in Fig. 4b). This, scientifically, doesn’t make sense, as the agreement will always be optimal, regardless of the quality of the model.
Secondly, Eq. (4) is not the Kramers-Kronig (K-K) relation; it is not even an integral transform. Probably, Eq. (4), can be derived from the K-K relation in some sense, but this either has to be shown explicitly or referenced clearly.
Finally, in the beginning of the manuscript, the complex modulus and the dynamic modulus (which is only the absolute value of the first) both seem to be denoted as E*, which severely hurts the transparency of Eqs. (1)-(7).
On a smaller note, in Figs. 9 and 10, showing only the (c) panel would be sufficient.
Comments on the Quality of English LanguageEnglish language is generally fine, except for minor weaknesses.
Author Response
I appreciate your suggestions and valuable advice. Based on the suggestions, I modified the manuscripts as follows.
Point 1: From a methodical point of view, I have a serious concern about the manuscript: If one performs experiments both to parametrize a model, and to test its predictions, the experimental data must be split into two sets: one for the determination of the model parameters, and one for testing the predictions of the parametrized model. In the manuscript, if I understand correctly, the same experimental data is used for fitting the parameters and to compare the model predictions with measured results (except only for the comparison in Fig. 4b). This, scientifically, doesn’t make sense, as the agreement will always be optimal, regardless of the quality of the model.
Response 1: The complex dynamic modulus test has two results, the dynamic modulus, and the corresponding phase angle, but a lot of historical data would only keep the dynamic modulus data, the phase angle data lost. For the asphalt mixture, if the phase angle data is lost, the phase angle data can be predicted by the phase angle master curve which is obtained from the corresponding dynamic modulus master curve by the K-K relation. The objective of this paper was to verify if the K-K relation applied to the PU mixture. In other words, if the phase angle data of the PU mixture is lost, whether the phase angle data be predicted from the corresponding dynamic modulus data by the K-K relation? Therefore, the test data should not be divided into two groups.
Point 2: Secondly, Eq. (4) is not the Kramers-Kronig (K-K) relation; it is not even an integral transform. Probably, Eq. (4), can be derived from the K-K relation in some sense, but this either has to be shown explicitly or referenced clearly.
Response 2: I am sorry about the confusion. The Eq. (4) is not the K-K relation or the integral transform. The Eq. (4) is the transformation process based on the K-K relation. The sentence was corrected in the new manuscript and the related reference was added.
Point 3: Finally, in the beginning of the manuscript, the complex modulus and the dynamic modulus (which is only the absolute value of the first) both seem to be denoted as E*, which severely hurts the transparency of Eqs. (1)-(7).
Response 3: All the dynamic modulus was changed into the absolute form, and the corrected form was revised in Eqs. (1)-(4).
Point 4: On a smaller note, in Figs. 9 and 10, showing only the (c) panel would be sufficient.
Response 4: The other two plots were deleted in Figs. 9 and 10 for better reading.
Point 5: English language is generally fine, except for minor weaknesses.
Response 5: I have revised the manuscript, and all the mistakes were corrected.
Thank you again for your hard, meticulous work!
Best regards!

Reviewer 4 Report
Comments and Suggestions for Authors
In this paper, Authors analyze the mechanical proprieties of the polyurethane mixture. The main tool used in this study is the Kramers-Kronig relation. However, in the paper lacks a suitable introduction and explanation of this relation and its importance and pertinence in the study performed. Therefore, a not expert in this field is completely disoriented by reading this paper starting from the introduction.
Furthermore, some technical problems must be fixed carefully.
- For instance, Eq. (3) should be a relation between real quantities. So, I wonder how this is possible if "i" in the denominator of the r.h.s. is the imaginary unity i^2=-1.
- How (7) is related to the definition of \delta(\omega) given in (5) and (6)?
- Eq. (6) is referred to (2) so that it should be " \delta ' " (\delta\prime) as well as "\alpha" should be "\alpha ' " (\alpha\prime)
Eqs. (8) and (9) are completely equivalent. It is enough to write just one of them (better Eq.(9)).
Eq. (11) is a relationship between log-functions, so the l.h.s. should be logE*.
The arguments of the log-function must be always dimensional! Therefore the label in several figures "frequency in Log(rad/s)" is wrong!
In some figures (i.e. fig. 5 or 6) a label ("d") is missing.
Overall, this paper may be of relevance to a reader interested in this topic but surely a significant improvement in the quality of its presentation is required before making further decisions.
English requires a revision by a native language expert in this field.
Author Response
I appreciate your suggestions and valuable advice. Based on the suggestions, I modified the manuscripts as follows.
Point 1: In this paper, Authors analyze the mechanical proprieties of the polyurethane mixture. The main tool used in this study is the Kramers-Kronig relation. However, in the paper lacks a suitable introduction and explanation of this relation and its importance and pertinence in the study performed. Therefore, a not expert in this field is completely disoriented by reading this paper starting from the introduction.
Response 1: I am sorry about the confusion in the introduction part, the Kramers-Kronig relation is not a widely spread concept. But the Kramers-Kronig relation is very interesting, it can be used to predict the phase angle data from the corresponding dynamic modulus data for the asphalt mixture. Therefore, if the phase angle data is lost or can not be obtained, we could obtain the phase angle data for the dynamic modulus data. Then, the properties of the asphalt mixture can be further analyzed.
The most important objective of the paper is to verify if the Kramers-Kronig relation can be applied to the PU mixture.
Point 2: For instance, Eq. (3) should be a relation between real quantities. So, I wonder how this is possible if "i" in the denominator of the r.h.s. is the imaginary unity i^2=-1.
Response 2: I am sorry about the mistakes in Eq. (3), and thank you for your advice. The equation was corrected in the new manuscript. The left part of Eq. (3) should be E*(ω) which represents the complex dynamic modulus. Then the complex dynamic modulus (E*) can be calculated based on the following equation:
E*=i*E’+E’’; E’=|E*|*sin(φ); E’’=|E*|*cos(φ);
The i represents the imaginary part, and the i2=-1.
Point 3: How (7) is related to the definition of \delta(\omega) given in (5) and (6)?
Response 3: The parameters in Eqs. (5), (6), and (7) were the same as those in Eqs. (1), (2), and (3), respectively.
Point 4: Eq. (6) is referred to (2) so that it should be " \delta ' " (\delta\prime) as well as "\alpha" should be "\alpha ' " (\alpha\prime)
Response 4: Thank you for your suggestion, the parameters in Eqs. (3), (6), and (7) were all corrected in different forms to avoid confusion. The rewritten equations were shown in the new manuscript.
Point 5: Eqs. (8) and (9) are completely equivalent. It is enough to write just one of them (better Eq.(9)).
Response 5: The Eq. (8) was deleted, and the original Eq. (9) was kept in the new manuscript.
Point 6: Eq. (11) is a relationship between log-functions, so the l.h.s. should be logE*.
Response 6: This equation is the sum of errors between the predicted and tested dynamic modulus in log form, therefore, the l.h.s only represents the sum of errors calculated according to the data under different loading frequencies and test temperatures. Then, the l.h.s should be represented in log form.
Point 7: The arguments of the log-function must be always dimensional! Therefore the label in several figures "frequency in Log(rad/s)" is wrong!
Response 7: All the mistakes were corrected in the new manuscript.
Point 8: In some figures (i.e. fig. 5 or 6) a label ("d") is missing.
Response 8: The label ("d") was added for each figure.
Thank you again for your hard, meticulous work!
Best regards!

Round 2
Reviewer 3 Report
Comments and Suggestions for Authors
Regarding Response 1:
If the research question is, whether the phase angle master curve can be retrieved without phase angle experimental data, only from the K-K relations, the answer, according to Fig. 4(b), is "no", and everything after Fig.(4) is irrelevant, because for the following models, the phase angle experimental data was used to fit the WLF-factors for the phase angle master curve.
As I was saying before; methodically, the manuscript has big flaws, because the authors can only achieve good agreement with experimental data, if they compare their models to the same data, which they used to parametrize the models.
That is not a test of the quality of the model; it is just a test, whether the model has enough free parameters to be fitted to the experimental data. And that is quite an unscientific question.
Regarding Response 3:
In Eq. (3), it is, in fact, not the absolute value, but the full complex dynamic modulus. That is why I raised the point in the first place.
Comments on the Quality of English LanguageLanguage is fine.
Author Response
I appreciate your suggestions and valuable advice. Based on the suggestions, I modified the manuscripts as follows.
Point 1: If the research question is, whether the phase angle master curve can be retrieved without phase angle experimental data, only from the K-K relations, the answer, according to Fig. 4(b), is "no", and everything after Fig.(4) is irrelevant, because for the following models, the phase angle experimental data was used to fit the WLF-factors for the phase angle master curve.
As I was saying before; methodically, the manuscript has big flaws, because the authors can only achieve good agreement with experimental data, if they compare their models to the same data, which they used to parametrize the models.
That is not a test of the quality of the model; it is just a test, whether the model has enough free parameters to be fitted to the experimental data. And that is quite an unscientific question.
Response 1: This question is about the concept or the objective of the paper. I am sorry about the confusion, and I will illustrate the concept of this paper in detail.
The equation (1) and (4) were taken as examples. The dynamic modulus data (|E*|) were obtained from the dynamic modulus test, then the dynamic modulus data (|E*|) can be fitted to get the corresponding master curve by the equation (1), the parameters group (δ, α, β, and γ) would be fitted at the same time. The fitted parameters group ( δ, α, β, and γ) was introduced into the equation (4) to produce the corresponding predicted phase angle data (φ). This process is the concept of the K-K relations.
If the dynamic modulus test data were changed, the parameters group (δ, α, β, and γ) for the equation (1) should be fitted again according to the new test data. In other words, the parameters group (δ, α, β, and γ) for dynamic modulus test data group 1 can not be introduced into equation (4) to predict the phase angle data for other dynamic modulus test data groups.
The objective of this paper is not to obtain a model with fixed parameters that could be used to predict all the phase angle data. Then the parameters for the model or equation (4) were not fixed, they should be fitted according to each dynamic modulus test data. For example, if there are two groups of dynamic modulus test data, equation (4) could be obtained for group 1 with fitted parameters group (δ, α, β, and γ), but we can not use the test data of group 2 to verify the precise of equation (4) obtained by the group 1.
The most important objective of this paper is to verify if the phase angle of the PU mixture could be obtained from the dynamic modulus test data according to the K-K relations, a model with fixed parameters can not be obtained to predict the phase angle data (φ) for all dynamic modulus test data groups. Therefore, this point is not the flaw of this paper.
Point 2: In Eq. (3), it is, in fact, not the absolute value, but the full complex dynamic modulus. That is why I raised the point in the first place.
Response 2: Based on the dynamic modulus test, we could obtain the dynamic modulus (|E*|) and phase angle (φ) with fixed values. Then the complex dynamic modulus (E*) can be calculated based on the following equation:
E*=i*E’+E’’; E’=|E*|*sin(φ); E’’=|E*|*cos(φ);
Therefore, the dynamic modulus (|E*|) is not the absolute value of the full complex dynamic modulus, the dynamic modulus is expressed in the form of |E*| only for easy reading based on the previous literature.
Thank you again for your hard, meticulous work!
Best regards!

Reviewer 4 Report
Comments and Suggestions for Authors
In this revised paper Authors have taken into account all my comments and suggestions. I have no objection to recommending this paper for publication in its current form.
Comments on the Quality of English LanguageEnglish requires a revision by a native language expert in this field.